# Prophylactic Awake Peripheral V-A ECMO during TAVR

**DOI:** 10.3390/jcm12030859

**Published:** 2023-01-20

**Authors:** Timur Lesbekov, Abdurashid Mussayev, Serik Alimbayev, Rymbay Kaliyev, Aidyn Kuanyshbek, Linar Faizov, Zhuldyz Nurmykhametova, Aigerim Kunakbayeva, Aigerim Sadykova

**Affiliations:** 1Department of Adult Cardiac Surgery, National Research Cardiac Surgery Center, Astana 020000, Kazakhstan; 2Department of Interventional Cardiology, National Research Cardiac Surgery Center, Astana 020000, Kazakhstan; 3Department of Perfusiology and Assisted Circulation Laboratory, National Research Cardiac Surgery Center, Astana 020000, Kazakhstan; 4Department of Anesthesiology, National Research Cardiac Surgery Center, Astana 020000, Kazakhstan

**Keywords:** transcatheter aortic valve replacement, extracorporeal membrane oxygenation, mechanical circulatory support, high-risk patient

## Abstract

Introduction: TAVR remains a complex procedure that may result in serious intraprocedural complications. In many of these circumstances, venoarterial extracorporeal membrane oxygenation (V-A ECMO) helps to manage complications, provides a hemodynamic back-up, and bridges to an emergency open heart surgery. The clinical outcomes of 27 patients who underwent prophylactic implantation of peripheral V-A ECMO (pV-A ECMO) during high-risk transcatheter aortic valve replacement (TAVR) cases are described. Methods: From June 2012 to October 2022, 590 consecutive patients underwent TAVR at our center. Of these, 27 patients (4.5%) underwent TAVR with pV-AECMO because they were deemed very high risk for periprocedural complications and formed the study population. Results: There were no pV-A ECMO, hemodynamic or TAVR implantation complications. Decannulation of the ECMO system was performed in 92.6% of cases at the end of the procedure in the hybrid-operating theatre. The mean duration of pV-A ECMO for procedure support was 51.4 ± 10.3 min. There were no ECMO-related vascular or bleeding complications. Conclusion: This study shows that the prophylactic placement of awake peripheral V-A ECMO provides excellent temporary cardio-circulatory and pulmonary support during very high-risk TAVR procedures.

## 1. Introduction

During high-risk transcatheter aortic valve replacement (TAVR) operations, there are several patient and/or procedure-related conditions that predispose to complications. Depressed left ventricular ejection fraction, severe pulmonary artery hypertension, and decompensated or high-grade heart failure constitute high-risk patients. In order to mitigate adverse outcomes in such cases, we implemented an algorithm to be used if such high-risk conditions were present. The strategy prefiguration was described by Raffa GM et al. [1] and summarized in an algorithm of prophylactic and emergency Veno-arterial extracorporeal membrane oxygenation (V-A ECMO) during TAVR for institutions without cardiac surgery on site. Our institution is the reference heart surgery and ECMO center in the country, so we decided to adapt the algorithm to our conditions (Figure 1). Clinical outcomes of 27 patients who underwent prophylactic implantation of peripheral V-A ECMO (pV-A ECMO) during high-risk TAVR cases are described.

## 2. Materials and Methods

From June 2012 to October 2022, 590 consecutive patients underwent TAVR at our center. Of these, 27 patients (4.5%) underwent TAVR with pV-A ECMO because they were deemed very high risk for periprocedural complications and formed the study population (Table 1). TAVR valves used included balloon-expandable Sapien XT™ (Edwards Lifesciences, Irvine, CA, USA), Myval™ (Meril Life Sciences Pvt. Ltd., Vapi, Gujarat, India), and self-expandable EvolutR™ (Medtronic, Irvine, CA, USA).

All TAVR operations and pV-A ECMO implantation (Stockert, Sorin Group Deutschland GMBH, München, Germany, and Deltasrtream MDC, Medos/Xenios, Xenios AG, Heilbronn, Germany) were performed via surgically assisted transfemoral approaches under monitored conscious sedation and local anaesthesia. The veno-arterial cannulation was performed via a mini cut-down open technique (Figure 2). Arterial and venous cannulae (Medtronic Bio-MedicusTM, Tijuana, Baja, California, Mexico) were selected according to the patient’s biometric parameters. Blood is drained from the right atrium and the inferior vena cava, oxygenated and decarboxylated in the ECMO device and returned to the iliac artery.

The pV-A ECMO was initiated in the operating theater prior to TAVR in all cases. The ECMO setting at the start of the procedure were: priming with 800 milliliters of balanced electrolyte isotonic solution (Sterofundin, B.BRAUN MELSUNGEN, AG, Germany). The initial pump speed was 1 L per minute. After vascular access is achieved, the circuit is connected to cannulae and flow is initiated at a low flow rate, increased incrementally to the target rate over a short time. Gas flow rates were set in relation to blood flow. The ECMO flow with a minimum of 1200 milliliters per minute and gas sweep rates were adjusted in each case to maintain target mean arterial pressure ≥ 70 mmHg and normocapnia, to maintain spontaneous breathing. During rapid pacing for balloon valvuloplasty, deployment and post-dilatation of the transcatheter heart valve, the pump was running stable speed.

## 3. Results

There were no pV-A ECMO, hemodynamic or TAVR implantation complications. Decannulation of the ECMO system was performed in 92.6% of cases at the end of the procedure in the hybrid-operating theatre. The mean duration of pV-A ECMO for procedure support was 51.4 ± 10.3 min. There were no ECMO-related vascular or bleeding complications (Table 2). There were no hospital mortality and morbidity in terms of bleeding, vascular complications, stroke, myocardial damage, renal insufficiency and thrombosis. Device success was achieved in all cases. Transprosthetic gradients during ECMO support were not measured due to a great risk of bias caused by mechanical circulatory support.

## 4. Discussion

Despite advancements in technology, TAVR remains a complex procedure that may result in serious intraprocedural complications, which can occur in up to 7.6% of TAVR operations [2]. In many of these circumstances, V-A ECMO can bridge the patients to emergency repair of structural damages by open heart surgery.

The use of Mechanical Circulatory Support (MCS) to manage complications and to provide a hemodynamic backup will increase the indications for TAVR [3,4].

Peripheral V-A ECMO is the most appropriate, short-term MCS device to support high-risk TAVR procedures [1]. From the technical standpoint, peripheral V-A ECMO only requires the insertion of arterial and venous cannulae over guidewires already present in the femoral vessels. In the scientific literature, there is a lack of differentiation between TAVR and ECMO-related vascular access complications. Despite non-specified data, vascular complications and bleeding occurred in a relatively small cohort of ECMO patients, reaching 16% [1]. In our series, all TAVR operations and pV-AE CMO implantation were performed via surgically assisted transfemoral approaches using a mini-cutdown open technique. There were no ECMO-related vascular or bleeding complications as vascular management was realized ad oculus. In all cases, the tunneling (through counerapertur) technique for cannulae insertion was used (Figure 2). This provides additional stability for cannulae and safe wound closure with minimal risk in case the prolongation of ECMO is needed. In addition, we found this maneuver much safer to control bleeding and other vascular complications when expandable introducers sheathe are used by snaring intraprocedurally, and vascular suturing at the end of the procedure for the TAVR site. Forthcoming decannulation is eased by pre-explored femoral vessels.

Among other MCS devices employed for emergency and elective TAVR procedures, cardiopulmonary bypass (CPB) has been adopted [5,6]. CPB is a feasible option intraprocedurally, but inconvenient and impractical for prolonged support. Other short-term MCS devices, such as Impella and Tandem Heart, have significant technical limitations that are time-consuming and challenging in TAVR procedures, especially in emergency situations [7].

There are currently concerns about the anesthesia management during high-risk TAVR, with the two main strategies—conscious monitored sedation and general anesthesia. The most common scenario for high-risk TAVR is general anesthesia, which was found the preferred strategy since it may assure a more well regulated cardiac and respiratory support [8,9]. In our institution, we found a conscious monitored sedation strategy more rational in the case of prophylactic ECMO institution. This approach carries good clinical outcomes providing (1) hemodynamic stability during balloon pre/post-dilatation; (2) hemodynamic stability during procedures in patients with high central venous pressure, and/or pulmonary capillary wedge pressure, mean pulmonary artery pressure (>50 mmHg) and low cardiac index; (3) hemodynamic stability while left ventricular recovery from rapid pacing; (4) reduces if there is need at all in high vasopressor or inotropes requirements; (5) protects concomitant high-risk intervention (coronary, valvular); (6) makes no rush for intubation in case of ventricular fibrillation; (7) prevents artificial ventilation. Moreover, the shortening of the length of stay may reduce costs [1].

The outcome of pV-A ECMO cases is favorable and patients are weaned off of ECMO support after the relief of aortic stenosis during the operation or the day after. The emergency cases are transferred to the operating theatre for the correction of structural damage (coronary ostia obstruction, annular, or ventricular rupture, aortic dissection, valve migration, apical disruption after transapical cases), which can occur in up to 7.6% of TAVR procedures [2,10]. The post-surgery course would most probably depend on the severity of intraprocedural complications. Post-cardiotomy V-A ECMO in these patients should be considered as a bridge to recovery. Unfortunately, results in this cohort show a much lower survival rate-about 60%. Surgical trauma, cardioplegic heart arrest, inotropic support and mechanical ventilation are additional factors influencing survival V-A ECMO.

In our opinion, open heart surgery should be performed on the same Catheter-Lab table in order to minimize the time for surgical restoration and the risk of multiorgan failure. It makes it crucial for the TAVR team to be composed of all of the experienced individuals: the referring cardiologist, the interventional cardiologist, the cardiac surgeon and anesthesiologist, the perfusionist, the scrub, and/or catheter-lab nurses.

### Limitations

The small number of patients and the retrospective nature of the analysis represent the major limitations of our study. It is clear that randomized trials are needed to address the feasibility and safety of pV-A ECMO for high-risk TAVR. Moreover, the influence of TAVR on late outcomes in patients with aortic stenosis at stage 3 and stage 4 of cardiac damage [11] warrants further investigation.

## 5. Conclusions

Prophylactic placement of awake peripheral V-A ECMO provides excellent temporary cardiocirculatory and pulmonary support during very high-risk TAVR procedures.

## Figures and Tables

**Figure 1 jcm-12-00859-f001:**
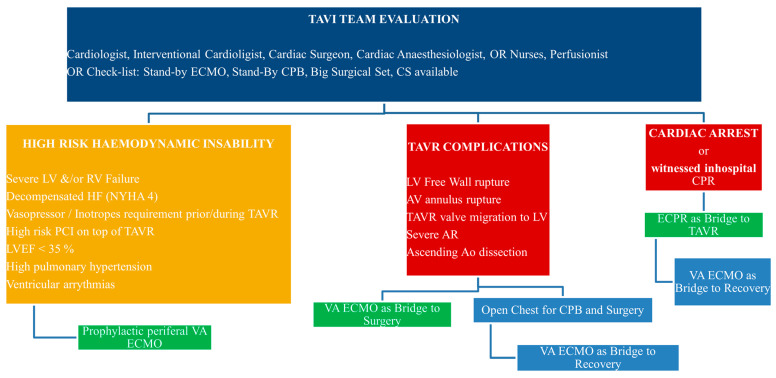
The algorithm of prophylactic and emergency veno-arterial extracorporeal membrane oxygenation for high-risk TAVR procedures with cardiac surgery on site. AV—Aortic Valve; CPB—Cardio Pulmonary Bypass; CS—Cardiac Surgeon; ECPR—Extracorporeal Cardio-Pulmonary Resuscitation; HF—Heart Failure; LVEF—Left Ventricular Ejection Fraction; NYHA—New-York Heart Association; OR—Operating Room; PCI—Percutaneous Intervention; RV—Right Ventricle; TAVR—Transcatheter Aortic Valve Replacement.

**Figure 2 jcm-12-00859-f002:**
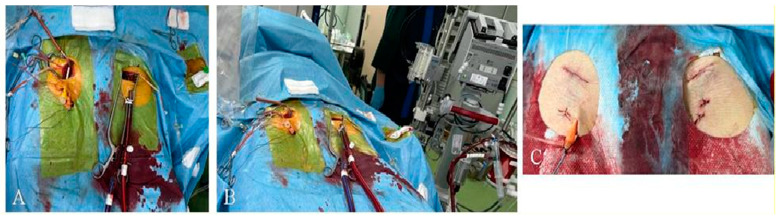
Mini-cutdown open technique for V-AECMO and TAVR cannulation. (**A**,**B**) intraprocedural view of running V-A ECMO; (**C**) final view of accesses for ECMO and TAVR.

**Table 1 jcm-12-00859-t001:** Baseline clinical characteristics.

Age (Years)	64.5 ± 6.4
Sex	
Male	22 (81.5%)
Female	5 (18.5%)
New York heart association functional class	
III	8 (29.6%)
IV	19 (70.4%)
Left ventricular ejection fraction (mean)	30.7 ± 14.1
<30% (*n*)	16 (66.7%)
Mean transvalvular gradient	36.25 ± 18.7
Invasive indexed aortic valve area (cm^2^/m^2^)	0.34 ± 0.17
Pulmonary artery pressure(mean)	56.1 ± 21
NT-pro BNP (mean)	11,003.4 ± 9827
Chronic obstructive pulmonary disease	7 (25.9%)
Diabetes mellitus	8 (29.6%)
Atrial fibrillation	8 (29.6%)
Previous Stroke	7 (25.9%)
Previous coronary artery disease	12 (44.4%)
Previous open heart surgery	4 (14.8%)
EuroScore II (%)	12.7% ± 8.8
Indications for TAVR	
Bicuspid aortic valve	17 (62.9%)
Tricuspid aortic valve	8 (29.6%)
Biological prosthesis failure (aortic + mitral)	1 (6.3%)
Valve-in-Ring	1 (6.3%)

**Table 2 jcm-12-00859-t002:** Prophylactic V-A ECMO characteristics and clinical course details.

Prophylactic V-A ECMO. *n* (%)	27 (100%)
Transfemoral surgical approach	
for TAVR	27 (100%)
for V-AECMO	27 (100%)
ECMO decannulation	
Periprocedural	25 (92.6%)
Post-operative day2	2 (7.4%)
Access site complication	0
Blood transfusion	0
ECMO time (mean, minutes)	51.4 ± 10.3
TAVR valve used	
EvolutR	10 (37%)
Myval	15 (55.5%)
Myval (aortic Valve-in-Valve + mitral Valve-in-Valve)	1 (3.7%)
Sapien XT	1 (3.7%)
Intensive care unit stay (days) (mean)	1.5 ± 0.8
Hospital stay (days) (mean)	4.6 ± 0.62
Mean transvalvular gradient	7.2 ± 3.2
Left ventricular ejection fraction at discharge day (mean)	42.6 ± 11.2
Pulmonary artery pressure at discharge day (mean)	33.1 ± 16
NT-pro BNP at discharge day (mean)	4249.5 ± 5017
Hospital Mortality	0
Morbidity	0

## Data Availability

The data presented in this study is available upon request from the corresponding author.

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
