# Peer review of "Prophylactic Awake Peripheral V-A ECMO during TAVR"

_jcm, 2023, doi:10.3390/jcm12030859_

Round 1

Reviewer 1 Report

I read with great interest and attention this paper by Dr. Lesbekov and Colleagues. The authors reported a series of 27 (4.6% of 590) consecutive (?) patients who underwent prophylactic implantation of peripheral venoarterial extracorporeal membrane oxygenation (V-A ECMO) during high-risk transcatheter aortic valve replacement (TAVR); in awake patients. These patients were "deemed very high risk for periprocedural complications". I think that the authors should be congratulated for their good results: 93.7% of decannulation; no ECMO-related complications.

I have several major and minor concerns with this paper.

Major concerns:

- The limited number of patients of the series (as still stated by the authors in the Limitations section)

- The lack of reported data as to the expected procedural risk according to universally recognized and accepted risk scores such as STS and EuroSCORE II.

- The lack of comments about reliability of hemodynamic evaluation (i.e. transprosthetic pressure gradients measurements) during ECMO assistance

- The lack of comments about any type of stand-by options. Did the authors considere the option of inserting only a vascular guide into the femoral vessels?

- The lack of clear informations as to the periprocedural ECMO setting.

Minor concerns:

- High pulmonary artery hypertension. It is preferable using <severe pulmonary hypertension>.

- The authors stated: Decannulation of the ECMO system was performed in 93.7% of cases. However, 26/27=96.3% and 25/27=92.6% !?

- The Abstract should be re-written. For instance, Introduction should include background information about the topic.

In conclusion, I cannot recommend publication of this manuscript in the current version. It have to undergo an extended revision as well as complete rewriting.

Author Response

- The limited number of patients of the series (as still stated by the authors in the Limitations section)

Answer. You are absolutely right - the number is small and so limits representativity and we stated it.

- The lack of reported data as to the expected procedural risk according to universally recognized and accepted risk scores such as STS and EuroSCORE II.

Answer. Risk scores such as STS and EuroSCORE II were really adapted for open surgery patients. Nowadays transcatheter valve technologies cover all - prohibitive, high and low risk patients. There are no standardised criteria for defining a “high-risk” procedure, there is general consensus due to a variable combination of clinical and anatomical factors. Among the first are the presence of a compromised functional class (NYHA III/IV), ventricular dysfunction, pulmonary hypertension, haemodynamic or electrical instability, heart failure despite optimised therapy, and the presence of comorbidities. So we tried to adapt clinical scenario (not surgical score) in the algorithm as preferable pathway for patient.

- The lack of comments about reliability of hemodynamic evaluation (i.e. transprosthetic pressure gradients measurements) during ECMO assistance

Answer. It is true - such evaluation should not be performed during ECMO assistance. So we did not also. Pre and post TAVR transvalvular pressure gradient measurements were done before and after ECMO assistance. In our next publication we will reveal this issue, but at the current we wanted to make the main focus on the algorithm and circulatory support.

- The lack of comments about any type of stand-by options. Did the authors considere the option of inserting only a vascular guide into the femoral vessels?

Answer. We did not use the option of inserting only a vascular guide into the femoral vessels. Stand by in our hands means that the whole heart team is present in cath-lab either for ECMO, or for CPB.

- The lack of clear informations as to the periprocedural ECMO setting.

Answer. The pV-A ECMO was initiated in the operating theater prior to TAVR in all cases. The ECMO flow and gas sweep rates were adjusted in each case to maintain target mean arterial pressure ≥ 70 mmHg and normocapnia, to maintain spontaneous breathing. During rapid pacing for balloon valvuloplasty, deployment and postdilatation of the transcatheter heart valve, the pump was running same stable speed.

Minor concerns:

- High pulmonary artery hypertension. It is preferable using <severe pulmonary hypertension>

Answer. We corrected this in the text

- The authors stated: Decannulation of the ECMO system was performed in 93.7% of cases. However, 26/27=96.3% and 25/27=92.6% !?

Answer. We corrected this in the text

- The Abstract should be re-written. For instance, Introduction should include background information about the topic.

Answer. We corrected this in the text

Reviewer 2 Report

In the present study Lesbekov and colleagues evaluate the use and outcome of periprocedural mechanical circulatory support (MCS) using ECMO in high-risk patients undergoing TAVI. They analyzed data from 27 patients in one center. 

Overall, they could show that patients with prophylactic hemodynamic support using ECMO provides a good periprocedural support in high-risk procedures, with low complication rate. 

With high numbers of TAVI procedures performed worldwide the topic is of interest. The manuscript is well written. 

However, I have some major concerns: 

- 62.9% received TAVI for bicuspid aortic valve disease. This is quite unusual and should be discussed. 

- Another limitation is the small sample size. There are several single center studies with small sample size. However, combining these data sets or showing a table with previously published studies including numbers, complications and outcome would at least provide an important overview. 

- I think it is absolutely essential to show follow-up data of these high-risk patients. Previous studies have shown mortality rates of 22% within 1 month and 39% 1-yaer mortality in patients undergoing TAVI with prophylactic ECMO (PMID: 30155732). 

Author Response

In the present study Lesbekov and colleagues evaluate the use and outcome of periprocedural mechanical circulatory support (MCS) using ECMO in high-risk patients undergoing TAVI. They analyzed data from 27 patients in one center. 

Overall, they could show that patients with prophylactic hemodynamic support using ECMO provides a good periprocedural support in high-risk procedures, with low complication rate. 

With high numbers of TAVI procedures performed worldwide the topic is of interest. The manuscript is well written. 

However, I have some major concerns: 

- 62.9% received TAVI for bicuspid aortic valve disease. This is quite unusual and should be discussed. 

Answer. Bicuspid aortic valve is the most common and frequent congenital heart disease. We supposed to attribute the most cases to bicuspid pathology by relatively young age (64) and combined visualization - echocardiography and computed tomography.

- Another limitation is the small sample size. There are several single center studies with small sample size. However, combining these data sets or showing a table with previously published studies including numbers, complications and outcome would at least provide an important overview. 

Answer. You are absolutely right - the number is small and so limits representativity and we stated it in the Limitations section. Certainly, it is a great idea to make a review article including all several single center studies in one. Thank you, we will do it.

- I think it is absolutely essential to show follow-up data of these high-risk patients. Previous studies have shown mortality rates of 22% within 1 month and 39% 1-yaer mortality in patients undergoing TAVI with prophylactic ECMO. 

Answer. Absolutely correct. The collaborative study from Germany (by Teresa Trenkwalder et al) demonstrated these disappointing results. They focused on using prophylactic ECMO for patients with depressed left ventricular ejection fraction. In our series there were patients with reduced left ventricular ejection fraction and relatively high transvalvular gradients (36.25±18.7) as an indirect marker of recoverability. In our study we wanted to make the main focus on the algorithm and circulatory support.  In our next publication that is in the progress, we will reveal this issue.

Round 2

Reviewer 1 Report

I think the authors responded elusively to the main objections I raised.

It is important to report EuroSCORE and/or STS risk score for each patient to help the reader understand the type of patient we are talking about.

There should be an ECMO setting at the start of the procedure, which is then adjusted according to the single patient's response.

The issue of the validity of transprosthetic gradients during ECMO must be at least mentioned, and not postponed to a future publication.

Minor issues

The acronym of "peripheral V-A ECMO" (i.e. pV-A ECMO) should be present even in the Abstract. 

In my opinion, "Prophylactic use of peripheral venous-arterial extracorporeal membrane oxygenation during complex transcatheter aortic valve replacement in awake patients" should be preferred to "Prophylactic Awake Peripheral V-A ECMO during TAVR".

I am sorry. It is impossible for me to recommend publication of the manuscript in the present form.

Author Response

It is important to report EuroSCORE and/or STS risk score for each patient to help the reader understand the type of patient we are talking about.

Answer. The information about EuroSCORE (12.7%±8.8) was added.

There should be an ECMO setting at the start of the procedure, which is then adjusted according to the single patient's response.

Answer. The information about an ECMO settingwas added to the main text. (Blood is drained from the right atrium and the inferior vena cava, oxygenated and decarboxylated in the ECMO device and returned to the iliac artery. The pV-A ECMO was initiated in the operating theater prior to TAVR in all cases. The ECMO setting at the start of the procedure were: priming with 800 milliliters of balanced electrolyte isotonic solution (Sterofundin, B.BRAUN MELSUNGEN, AG, Germany). The initial pump speed was 1 liter per minute. After vascular access is achieved, the circuit is connected to cannulae and flow is initiated at a low-flow rate, increased incrementally to the target rate over a short time. Gas flow rates were set in relation with blood flow. The ECMO flow with a minimum of 1200 milliliters per minute and gas sweep rates were adjusted in each case to maintain target mean arterial pressure ≥ 70 mmHg and normocapnia, to maintain spontaneous breathing. During rapid pacing for balloon valvuloplasty, deployment and postdilatation of the transcatheter heart valve, the pump was running stable speed.)

The issue of the validity of transprosthetic gradients during ECMO must be at least mentioned, and not postponed to a future publication.

Answer. The information was added to the main text. (Transprosthetic gradients during ECMO support were not measured due to great risk of bias caused by mechanical circulatory support.)

Minor issues

The acronym of "peripheral V-A ECMO" (i.e. pV-A ECMO) should be present even in the Abstract. 

Answer. The acronym of "peripheral V-A ECMO" (i.e. pV-A ECMO) was added to the Abstract

In my opinion, "Prophylactic use of peripheral venous-arterial extracorporeal membrane oxygenation during complex transcatheter aortic valve replacement in awake patients" should be preferred to "Prophylactic Awake Peripheral V-A ECMO during TAVR".

Answer. Please, we would like to save the original name of the paper.

Reviewer 2 Report

Unfortunately, I cannot see any changes that have been made within the manuscript. 

Author Response

Dear Reviewer,

Please, find the uploaded manuscript with changes.

Thank you very much.